# ArtAug: Iterative Enhancement of Text-to-Image Models via Synthesis–Understanding Interaction

## Abstract

The emergence of diffusion models has significantly advanced image synthesis. Recent studies of model interaction and self-corrective reasoning approaches in large language models offer new insights for enhancing text-to-image models. Inspired by these studies, we propose a novel method called `ArtAug` for enhancing text-to-image models via model interactions with understanding models. In the interactions, we leverage human preferences implicitly learned by image understanding models to provide fine-grained suggestions for image generation models. The interactions can modify the image content to make it aesthetically pleasing, such as adjusting exposure, changing shooting angles, and adding atmospheric effects. The enhancements brought by the interaction are iteratively fused into the generation model itself through an additional enhancement module. This enables the generation model to produce aesthetically pleasing images directly with no additional inference cost. In the experiments, we verify the effectiveness of `ArtAug` on advanced models such as FLUX, Stable Diffusion 3.5 and Qwen2-VL, with extensive evaluations in metrics of image quality, human evaluation, and ethics. The source code and models will be released publicly.

## 1 Introduction

Diffusion models (Sohl-Dickstein et al., 2015; Ho et al., 2020) have been extensively studied in recent years. With the development of large-scale image datasets (Schuhmann et al., 2022; Gu et al., 2022), large text-to-image models (Rombach et al., 2022; Chen et al., 2023; Saharia et al., 2022) have rapidly developed and demonstrated strong application potential. Downstream tasks such as interactive creation (Liu et al., 2024c), controllable image generation (Zhang et al., 2023), and consistent story generation (Zhou et al., 2024) all require the generated content to align with human preferences. However, pre-trained text-to-image models often struggle to produce satisfactory images without high-quality training datasets or human guidance tailored for specific cases.

To guide image generation models in producing high-quality images, current research primarily focuses on three aspects: **1) Data refinement** (Chen et al., 2024a; Schuhmann et al., 2022) are employed to eliminate low-quality images from large training datasets, thereby preventing them from negatively impacting the model's performance. **2) Prompt engineering** (Wang et al., 2024b; Cao et al., 2023) aims to craft detailed prompts to guide the model in producing superior-quality images. **3) Alignment training** (Wallace et al., 2024; Fan et al., 2024) focuses on aligning the model's generative inclinations with human preferences via training. However, these methods all have certain limitations. Data refinement can only be used for coarse filtering. Directly filtering out low-quality images requires meticulous efforts and potentially leads to overfitting due to the insufficient amount of data. Prompt engineering based on language models might result in generated images containing content that is inconsistent with the user-provided prompts, thereby compromising the text-image correlation. Alignment training is currently the key method for improving image quality. The mainstream alignment training methods, including Reinforcement Learning from Human Feedback (RLHF) (Ouyang et al., 2022) and Direct Preference Optimization (DPO) (Rafailov et al., 2024), require a large amount of manually annotated data, leading to extremely high costs.

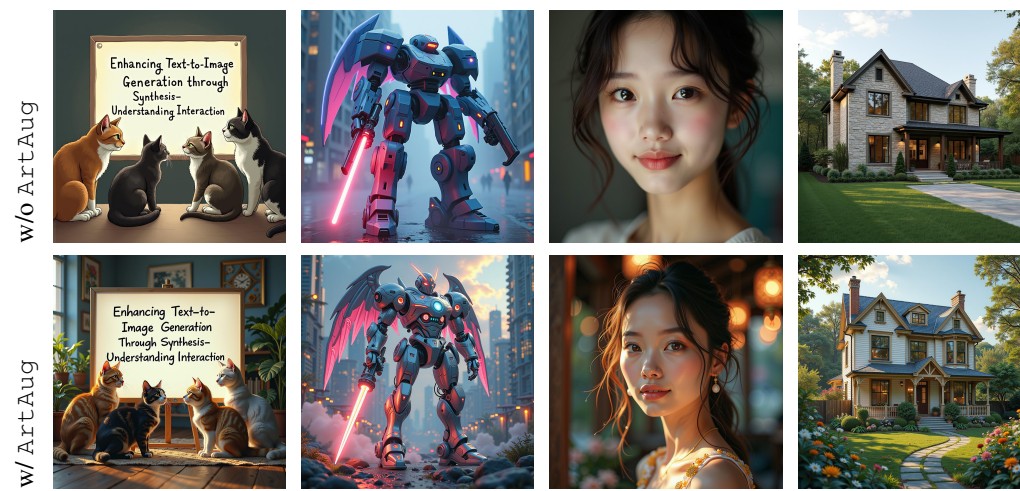

Figure 1: Image examples improved by `ArtAug`. The base text-to-image model is FLUX.1[dev].

On the other hand, recent studies of model interaction and self-corrective reasoning provide us with new insights for enhancing the capabilities of image generation models. Particularly, GPT-o1 (OpenAI, 2024) significantly enhances the capabilities of LLMs (Large Language Models) (Brown, 2020) through self-corrective reasoning via the model itself, at the expense of longer computation time. LLMs are trained on human-generated data and potentially understand human interpretations and preferences for aesthetics. Recent studies have preliminarily demonstrated the feasibility of guiding image generation models through interactive conversations using language models (Huang et al., 2024). Some multimodal models (Wang et al., 2024a; Chen et al., 2024b; Liu et al., 2024b) are capable of understanding image content and expressing it through natural language, motivating us to explore the deeper assistive roles of LLMs in relation to image generation models.

To address the current challenges faced by image generation models, inspired by the model interaction and self-corrective reasoning approaches, we propose a novel text-to-image generation model enhancement approach called `ArtAug`. As shown in Figure 1, `ArtAug` can significantly improve the image quality, aligning the generated image content with human preference. Our framework, `ArtAug`, introduces a paradigm shift by replacing the human annotator with a highly capable MLLM-based "AI Art Director". While the training workflow is multi-staged, it is computationally manageable, offering a scalable and cost-effective alternative to human-in-the-loop alignment.

The framework `ArtAug` is presented in Figure 2. There are three modules in `ArtAug`, including a **generation module** for text-to-image generation, an **understanding module** for analyzing and refining the image content, and an **enhancement module** for improving the generation module. Firstly, we design an interactive image synthesis algorithm, where the understanding module provides fine-grained modifications for the generation module to produce enhanced images. Secondly, we build a pairwise training dataset by generating and filtering image pairs. Thirdly, we introduce differential training to teach the enhancement module to capture differences between original and enhanced images. Fourthly, we integrate the enhancement module into the generation module, imbuing the generation module with the enhancement capability brought by interactions, without extra computational cost. This process is iterated to progressively improve generation. In experiments, we train the enhancement module on advanced text-to-image models, including FLUX.1[dev] (Labs, 2024) and Stable Diffusion 3.5 (Esser et al., 2024). `ArtAug` `ArtAug` significantly improves image quality, generating more aesthetically pleasing results, evidenced by various evaluation metrics. We will release the source code and models. Overall, the contributions of this paper include:

- We design an interaction algorithm between a generation module and an understanding model in image synthesis, demonstrating that current multimodal LLMs can guide text-to-image models to generate high-quality images aligned with human preferences.

- We propose `ArtAug`, a framework for improving text-to-image models. By learning the differences between images before and after interaction, we iteratively enhance the capabilities of the text-to-image model.

- We train the `ArtAug` enhancement module based on advanced text-to-image models. Extensive experiments consistently demonstrate the effectiveness of `ArtAug` in improving image quality across multiple aspects.

## 2 RELATED WORK

### 2.1 LARGE IMAGE SYNTHESIS MODELS

In recent years, diffusion models (Sohl-Dickstein et al., 2015; Ho et al., 2020) have achieved significant breakthroughs in the field of image synthesis, even reaching the level of human artists. Since the introduction of Latent Diffusion (Rombach et al., 2022), large diffusion models pre-trained on large-scale text-image datasets (Schuhmann et al., 2022; Lin et al., 2014; Gu et al., 2022) have made considerable advancements. The generative capabilities of these models have been steadily improved, including both UNet-based models (Ronneberger et al., 2015; Rombach et al., 2022; Podell et al., 2023; Sauer et al., 2025) and the more recent DiT-based models (Li et al., 2024; Chen et al., 2023; Esser et al., 2024; Labs, 2024). Notably, DiT (Diffusion Transformer) (Peebles & Xie, 2023) has considerably enhanced both the convergence speed and the generalization ability of image generation models, establishing itself as one of the most popular architectures in the realm of image synthesis. To further enhance image quality in terms of text-image alignment and aesthetic appeal, various approaches, such as data refinement (Chen et al., 2024a; Schuhmann et al., 2022), prompt engineering (Wang et al., 2024b; Cao et al., 2023), and alignment training (Wallace et al., 2024; Fan et al., 2024), have been extensively investigated. Inspired by these studies, we propose a new approach, which synthesizes data via model interactions and iteratively improves the text-to-image model after data filtering.

### 2.2 ALIGNING MODELS WITH HUMAN PREFERENCES

Text-to-image models pre-trained on extensive text-image datasets have demonstrated rudimentary image generation abilities, but these models often produce suboptimal quality images without fine-tuning (Liu et al., 2024a). Currently, alignment training stands as the principal method for improving image quality by aligning generated content with human preferences. Alignment training is initially investigated in large language models (Ouyang et al., 2022; Rafailov et al., 2024), and has recently been applied to diffusion models. For example, based on reinforcement learning, approaches like DPOK (Fan et al., 2024) and DDPO (Black et al., 2023) gather human preferences on model-generated outputs for fine-tuning text-to-image models. Similarly, Diffusion-DPO (Wallace et al., 2024) and SPO (Liang et al., 2024) employ auxiliary models to model human preferences, using DPO (Rafailov et al., 2024) to fine-tune diffusion models accordingly. However, because human preferences are difficult to quantify, these alignment training methodologies necessitate extensive manually labeled datasets, which are prohibitively expensive to produce. Inspired by these studies, we explore the possibility of using multimodal LLMs to replace manual annotation, aiming to obtain a large amount of training data at a lower cost for alignment training.

## 3 METHODOLOGY

The framework of `ArtAug` is presented in Figure 2. `ArtAug` consists of three key steps: interactive image synthesis, dataset construction and differential training. The three steps are applied to the model iteratively. In this section, we provide a detailed description of each step.

### 3.1 INTERACTIVE IMAGE SYNTHESIS

Text-to-image models usually tend to generate simple content when given simple prompts. Prompt engineering is generally essential for generating high-quality images, but manually crafting high-quality prompts requires the expertise of human experts. The image generation module (a text-to-image model) itself struggles to generate detailed and aesthetically pleasing images. To address this challenge, we propose an interactive algorithm and utilize an additional understanding module (a multimodal LLM) to aid the generation module.

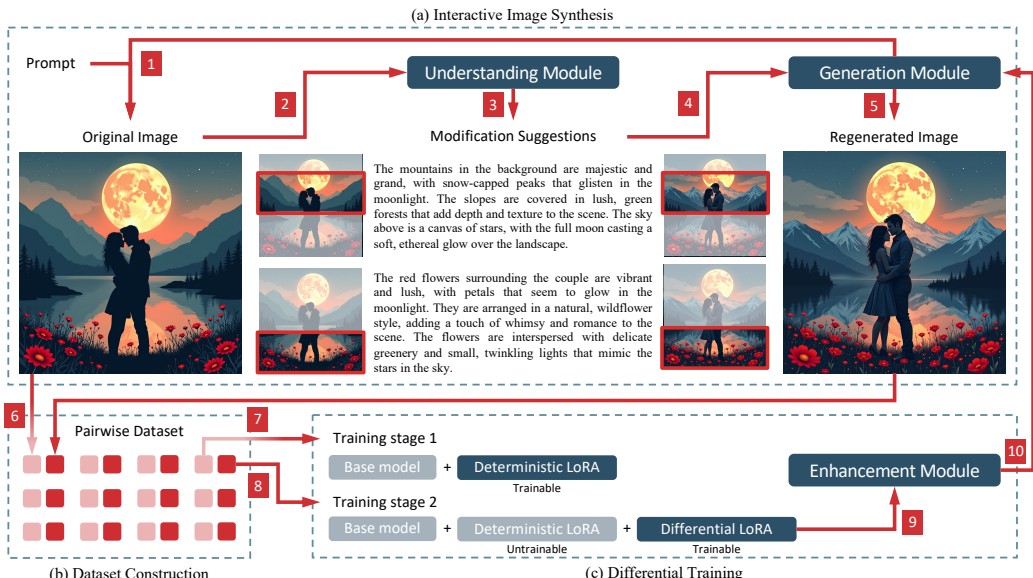

Figure 2: The framework of `ArtAug` encompasses three key steps. **(a) Interactive image synthesis**: leverage the generative module to create high-quality images, aided by the understanding module. **(b) Dataset construction**: generate a large amount of image pairs and filter them to build a dataset. **(c) Differential training**: train the enhancement module to optimize the performance of the generation module. This enhancement process can be iteratively applied to the model, facilitating iterative improvement.

The interactive algorithm includes three steps: generation, understanding, and refinement. First, we use the original generation pipeline of the text-to-image model to generate an image $X$. Second, we employ the understanding module $u$ to analyze the image content and generate modification suggestions. The understanding module is implemented based on multimodal LLMs due to their significant image understanding and grounding capabilities. The modification suggestions provided by the understanding module are in the form of $n$ pairs of prompt and bounding box, which are formulated as $u(X) = \{(P_i, \boldsymbol{M}_i)\}_{i=1}^n$, where the bounding box $\boldsymbol{M}_i \in \{0,1\}^{H \times W}$ represents the location and the prompt $P_i$ describes the corresponding modified content. To improve computational efficiency, we directly generate all bounding boxes and prompts through a single-turn dialogue. Third, we use the generation module to regenerate the image according to the suggestions for image modifications. To finely control the content of images and ensure that each prompt affects the corresponding area, we design a partitioned image generation method based on previous studies (Li et al., 2023; Bar-Tal et al., 2023). Assuming that the original model output is $\hat{\boldsymbol{\epsilon}}_\theta(P, t, \boldsymbol{h})$, where $P$ is the original prompt, $t$ is the timestep of the denoising process, and $\boldsymbol{h} \in \mathbb{R}^{H \times W}$ is the latent representation of the image, we use the weighted average of $\{\hat{\boldsymbol{\epsilon}}_\theta(P_i, t, \boldsymbol{h})\}_{i=1}^n$ to replace the original model output. The pseudo code of the interactive algorithm is presented in Appendix A.1. Intuitively, this algorithm employs the model to infer according to different prompts and then performs a weighted average of the results based on the location information, where the weight denotes the region of each prompt.

Unlike directly refining the prompt words, this interactive algorithm can preserve the basic composition of the image and ensure consistency in visual content, making the image pairs more suitable for subsequent differential training. We provded several comparison examples in Appendix A.2.

## 3.2 DATASET CONSTRUCTION

While the procedures outlined in the previous subsection can enhance the quality of generated images without the need for additional training, this algorithm presents two notable drawbacks. 1) Slow computational speed. The fine-grained prompts necessitate independent forward inferences through the model, leading to an increase in overall computation time to $n + 1$ times the original time. Consequently, this extends the generation process to several minutes. 2) Risk of bad cases.

Due to the lack of end-to-end training in the modified image generation pipeline, there is a potential for producing unrealistic images with higher probability. To overcome these drawbacks, we aim to consolidate the enhancements produced through interaction into the text-to-image model itself via post-training. Specifically, we generate a large batch of image pairs $(X, X')$ using the interactive image synthesis algorithm, where $X$ represents images generated by the original text-to-image pipeline and $X'$ represents the interactively refined images. We then filter these image pairs to construct a high-quality dataset for training.

In the data filtering process, we apply stringent filtering criteria. Fundamentally, we expect that images enhanced through interaction should be more aesthetically pleasing than the originals. Therefore, we only retain image pairs with increased aesthetic scores (Schuhmann et al., 2022). Additionally, we need to ensure that the enhanced images are consistent with the semantic meaning of the text prompts. To achieve this, we use the CLIP model (Radford et al., 2021) to filter out all image pairs where the text-image similarity decreases. These two filtering steps can effectively eliminate a significant portion of the data unsuitable for training. However, given that the prompts are collected from real users and exhibit complex content, we conduct further meticulous manual reviews. During the manual review, we remove the content related to pornography, violence, politics, and racial discrimination, and ensure that the enhanced images in the image pairs show a clear improvement in quality. Appendix A.3 shows several categories of the final retained training data, where the image pairs exhibit significant improvements in various aspects, including lightning, detail, composition, ambiance, clarity, and color. Beyond these fundamental aesthetic enhancements, our algorithm can achieve more advanced effects, including particle effects, shooting angle, exposure compensation, style adjustment, background blur and color gradient, etc. This shows our method is not just a generic "make it prettier" filter, but is actively performing targeted, compositional improvements guided by the MLLM's understanding.

### 3.3 DIFFERENTIAL TRAINING

After filtering and review, approximately 1% to 2% images are retained. The amount of data is relatively small, and directly using these data to fine-tune the model leads to overfitting, as shown in our preliminary experiments. Based on the observation, we propose a differential training approach to learn the differences between images, rather than directly learning the images enhanced by interactions.

The model structure of the enhancement module is LoRA (Low-Rank Adaptation) (Hu et al., 2021). Assume that the parameters of fully connected layers in the original model are formulated as

$$\theta = \{\boldsymbol{W}_i\}_{i=1}^m. \tag{1}$$

For each parameter matrix $\boldsymbol{W}_i \in \mathbb{R}^{d_1 \times d_2}$, the parameter matrix after adding LoRA becomes $\boldsymbol{W}_i + \boldsymbol{B}_i \boldsymbol{A}_i$, where $\boldsymbol{A}_i \in \mathbb{R}^{r \times d_2}$ and $\boldsymbol{B}_i \in \mathbb{R}^{d_1 \times r}$ are two low-rank matrices. The LoRA rank $r$ is a hyperparameter. We use $\phi = \{(\boldsymbol{A}_i, \boldsymbol{B}_i)\}_{i=1}^m$ to denote all LoRA parameters and use "$\oplus$" to denote the operator of adding LoRA parameters, i.e.,

$$\theta \oplus \phi = \{\boldsymbol{W}_i + \boldsymbol{B}_i \boldsymbol{A}_i\}_{i=1}^m. \tag{2}$$

We model the standard diffusion model training objective for a single image $X$ with prompt $P$ as minimizing the loss $\mathcal{L}_{\mathrm{DM}}$:

$$\min_\phi \mathbb{E}_{t,\boldsymbol{\epsilon}} \left[ \mathcal{L}_{\mathrm{DM}}(\theta \oplus \phi, P, X, t, \boldsymbol{\epsilon}) \right], \tag{3}$$

where $\phi$ represents the trainable LoRA parameters and $\theta$ are the frozen base model weights. The specific form of $\mathcal{L}_{\mathrm{DM}}$ depends on the diffusion formulation (e.g., DDPM (Ho et al., 2020) or Flow Matching (Esser et al., 2024)).

To learn the enhancement from an image pair $(X, X')$, where $X$ is the original and $X'$ is the enhanced version, our differential training proceeds in two conceptual steps for each pair:

1. **Deterministic LoRA ($\phi_1$):** First, we train a LoRA model $\phi_1$ to perfectly reconstruct the original image $X$. This anchors the model to the base content.

$$\phi_1 = \arg\min_\phi \mathbb{E}_{t,\boldsymbol{\epsilon}} \left[ \mathcal{L}_{\mathrm{DM}}(\theta \oplus \phi, P, X, t, \boldsymbol{\epsilon}) \right]. \tag{4}$$

2. **Differential LoRA ($\phi_2$):** Second, we freeze the base model $\theta$ and the deterministic LoRA $\phi_1$, and train a new LoRA, $\phi_2$, to fit the enhanced image $X'$. This forces $\phi_2$ to capture only the delta between $X$ and $X'$.

$$\phi_2 = \arg\min_{\phi} \mathbb{E}_{t,\epsilon} \left[ \mathcal{L}_{\text{DM}}(\theta \oplus \phi_1 \oplus \phi, P, X', t, \epsilon) \right]. \tag{5}$$

The final enhancement module for this pair is $\phi_2$, which represents the learned aesthetic transformation. We discard $\phi_1$. This process is repeated for all filtered pairs in our dataset, yielding a set of differential LoRA modules.

### 3.4 ITERATIVE IMPROVEMENT

Through differential training, we obtain a LoRA model $\phi$ that can enhance the generative capabilities of the text-to-image model. This LoRA model can be fused into the base model, i.e., let

$$\theta \leftarrow \theta \oplus \alpha \Phi \big( \theta \oplus \Phi(\theta, X), X' \big), \tag{6}$$

where $\alpha$ is the weight of the LoRA. To enhance the stability of the model's preferences, we average the LoRA parameters across multiple image pairs. Based on this iterative formula, we make it possible to continue generating data through the interaction processes. Consequently, the data generation and the differential training process can be iteratively repeated until the interactive algorithm can no longer significantly improve the quality of the generated images. Ultimately, we obtain a series of stacked LoRA models $\{\phi^{[1]}, \phi^{[2]}, \dots\}$. We merge them into a single LoRA model by concatenating the corresponding matrices. The use of the entire enhancement module is consistent with that of a standard LoRA model and maintains compatibility with other LoRA models. Additionally, users can adjust the influence of the enhancement module on the text-to-image model by tuning the weight of the merged LoRA model, thereby achieving controllable generation.

From another perspective, this iterative enhancement process involves updating the model parameters at each iteration, akin to a gradient descent step. We provide a detailed empirical analysis of the iterations in Section 4.2. The trainable LoRA parameters correspond to the gradient. The parameter $\alpha$ corresponds to the learning rate in gradient descent. A smaller $\alpha$ can make the training process more stable, but it will slow down the convergence speed. The number of averaged LoRA models corresponds to the batch size. In this manner, we can employ human preference, an inherently non-differentiable training objective, for the training of the model implicitly via data synthesis.

## 4 EXPERIMENTS

We conduct extensive experiments to demonstrate `ArtAug`'s effectiveness, including improving off-the-shelf models and thoroughly investigating each component. Aesthetics is a complex and subjective concept; therefore, we adopt a diversified evaluation approach, including basic image quality metrics, human preference models, and human evaluation.

### 4.1 IMPROVING OFF-THE-SHELF MODELS

#### 4.1.1 EXPERIMENTAL SETTINGS

We train the `ArtAug` enhancement module based on the advanced text-to-image models FLUX.1[dev] (Labs, 2024) and Stable Diffusion 3.5 (Esser et al., 2024). In the interaction algorithm, the understanding module is implemented based on Qwen2-VL-72B (Wang et al., 2024a) due to its sufficiently accurate visual grounding capabilities, which enable the generation of fine-grained bounding boxes and prompts. The prompt used in Qwen2-VL-72B is presented in Appendix B.1. We provide detailed discussions about the selection of the multimodal LLM in Appendix B.2, including a comparative analysis between six SOTA multimodal LLMs. Our experiments do not require a text-image dataset; we only use a dataset of prompts. In each training iteration, we randomly sample approximately 3k prompts from the DiffusionDB dataset (Wang et al., 2022). Considering that these prompts are collected from users on the internet and some may contain ambiguous semantics, we refine the prompts using Qwen2-VL-72B before generating images. After filtering and reviewing as described in Section 3.2, we engaged two human annotators for lightweight data filtering due to

Table 1: Quantitative results on basic image quality.

|  | Aesthetic ↑ | CLIP ↑ |
|---|---|---|
| FLUX.1[dev] | 6.35±0.005 | 26.92±0.046 |
| FLUX.1[dev] + `ArtAug` | **6.81±0.005** | **26.97±0.048** |
| Stable Diffusion 3.5 | 6.12±0.005 | 27.52±0.043 |
| Stable Diffusion 3.5 + `ArtAug` | **6.61±0.004** | **28.15±0.044** |

Table 2: Quantitative results on preference models.

|  | PickScore ↑ | MPS ↑ | HPS ↑ | ImageReward ↑ |
|---|---|---|---|---|
| FLUX.1[dev] | 42.22 | 47.52 | 49.36 | 48.21 |
| FLUX.1[dev] + `ArtAug` | **57.78** | **52.48** | **50.64** | **51.79** |
| Stable Diffusion 3.5 | 40.97 | 44.94 | 49.35 | 44.53 |
| Stable Diffusion 3.5 + `ArtAug` | **59.03** | **55.06** | **50.65** | **55.47** |

concerns about harmful content. We trained a differential LoRA model for each image pair. The learning rate is set to $1 \times 10^{-4}$, with a batch size of 1, and the LoRA model is trained for 400 steps. The LoRA rank is manually adjusted to 4, 8, or 16 to ensure convergence on the training image. The loss function is consistent with the flow match theory (Esser et al., 2024), and other training hyperparameters are consistent with those of the base text-to-image model itself. One full iteration of `ArtAug`, including generating 5k initial pairs, MLLM-based refinement, filtering, and training all differential LoRAs, was completed under 24 hours on a single 8×A100 node. The human involvement was limited to approximately 2 person-hours for final data review per iteration, a fraction of the cost of typical RLHF campaigns.

### 4.1.2 QUANTITATIVE COMPARISON

After training, we randomly sample 10k prompts in DiffusionDB (Wang et al., 2022) to evaluate the quality of the generated images. These prompts are not used in the data generation. The evaluation metrics include two categories. 1) **Basic image quality metrics**: Aesthetic (Schuhmann et al., 2022) and CLIP (Radford et al., 2021), which are used to measure the aesthetic quality of images and the alignment between text and image, respectively. 2) **Preference models**: PickScore (Kirstain et al., 2023), MPS (Zhang et al., 2024), HPS (Wu et al., 2023), and ImageReward (Xu et al., 2023). These models are classifier models trained on human-annotated data, and their results can be regarded as approximations of human preferences.

The quantitative results are presented in Table 1 and Table 2. On the basic aesthetic metric, the model trained with `ArtAug` demonstrates significant improvement. This clearly indicates the efficacy of `ArtAug` in enhancing image quality. Furthermore, the CLIP text-image similarity metric does not exhibit a noticeable decline, suggesting that `ArtAug` does not compromise the original text comprehension ability of the base model. In Table 2, all preference models consistently demonstrate the effectiveness of `ArtAug`. Overall, `ArtAug` is capable of enhancing the fundamental capabilities of the text-to-image model. Notably, the strong results on FLUX.1[dev] show that `ArtAug` delivers significant gains even on highly optimized models. While DPO/RLHF optimize for text-image alignment and general human preference, `ArtAug` adds orthogonal gains by using VLMs' region-aware understanding to boost aesthetics and visual refinement.

### 4.1.3 HUMAN EVALUATION

Some studies (Podell et al., 2023; Jiang et al., 2024) have highlighted the limitations of automatically computed evaluation metrics, prompting us to conduct an additional double-blind human evaluation. We invite 20 participants to take part in this evaluation. In each round, participants are shown two images: one generated by the original text-to-image model and the other generated by the `ArtAug`-enhanced model. Similar to GenAI-Arena (Jiang et al., 2024), the positions of the two images are randomized. The instructions provided to participants are presented in Appendix C.

Table 3: Quantitative results on human evaluation.

|  | w/o `ArtAug` is better | Tie | w/ `ArtAug` is better |
|---|---|---|---|
| FLUX.1[dev] | 39.18% | 14.89% | **45.93%** |
| Stable Diffusion 3.5 | 39.66% | 9.49% | **50.85%** |

Table 4: Quantitative results on ethics, evaluated using NudeNet.

|  | w/o `ArtAug` | w/ `ArtAug` |
|---|---|---|
| FLUX.1[dev] | 6.19 | **4.44** |
| Stable Diffusion 3.5 | 4.97 | **2.88** |

Each participant is asked to select the image with better visual quality or to choose "tie". We record the percentage of user votes, as shown in Table 3. `ArtAug` achieves winning rate of 45.93% and 50.85%, demonstrating the effectiveness of `ArtAug` in enhancing visual quality.

### 4.1.4 ETHICS CONSIDERATION

Although `ArtAug` enhances image generation capabilities, it also raises ethical concerns. We find that user prompts sourced online often include harmful content (e.g., pornography, violence). Moreover, pre-trained image understanding models may encode biases toward such content, leading the interaction algorithm to produce suggestive imagery. This requires human moderation during `ArtAug`'s iterative training to prevent bias propagation. To mitigate this, we leverage NudeNet (Bedapudi, 2019) to measure harmfulness scores of generated images. As shown in Table 4, `ArtAug` does not increase the model's propensity to generate harmful content.

### 4.1.5 PROMPT-FOLLOWING EVALUIATION

Whether the original prompt-following capability of models would be compromised when aligning to human preferences remains a critical issue. Studies (Huang et al., 2025; Zhang et al., 2024) have highlighted the limitations of CLIP scores in evaluating prompt-following performance. Therefore, we benchmarked our trained model against other publicly available models on T2I-CompBench++ (Huang et al., 2025), an evaluation framework for text-to-image generation. The results are presented in Table 5. Although `ArtAug` is not explicitly optimized for prompt-following capabilities, we observed slight improvements in this aspect. The reason is that the image understanding model can interpret both visual content and text content to provide modification suggestions. While this approach occasionally incorporates aesthetic details beyond prompt specifications (which may reduce CLIP scores), the fine-grained evaluation benchmark T2I-CompBench++ demonstrates that generated content does not violate fundamental instruction constraints. Consequently, `ArtAug` achieves slight enhancement in prompt-following capabilities based on the advanced model.

### 4.2 IMPACT OF ITERATION STEP

To better understand the changes in the model's capabilities throughout the iterative training process, we analyzed the data of image pairs generated in each iteration. Some statistical indicators are presented in Figure 3. In each iteration, our primary focus is on the enhancement of image aesthetics by the interaction algorithm. It can be observed from the figure that the aesthetic scores consistently improve after interaction. This enhancement capability is ingrained into the model during training and carries over to the next iteration, enabling continuous improvement. We also calculated the correlation between images and prompts before and after interaction using the CLIP model. It should be noted that these prompts are refined by the language model, so the CLIP scores appear slightly higher than those in Table 1. Although the interaction algorithm may sometimes alter the image content away from its original semantics, especially when the prompt contains terms like ugly, dirty, or bloody, our rigorous data filtering process eliminates such data to prevent compromising the model's original capabilities. Additionally, we calculated the cosine similarity of images before and after interaction using the vision encoder component (Dosovitskiy, 2020) of the CLIP model. As

Table 5: Prompt-following evaluation on T2I-CompBench++.

|  | Color ↑ | Shape ↑ | Texture ↑ | Overall ↑ |
|---|---|---|---|---|
| Stable Diffusion v1.4 | 0.3765 | 0.3576 | 0.4156 | 0.3832 |
| Stable Diffusion v2 | 0.5065 | 0.4221 | 0.4922 | 0.4736 |
| Stable Diffusion XL | 0.5879 | 0.4687 | 0.5299 | 0.5288 |
| Pixart-$\alpha$-ft | 0.6690 | 0.4927 | 0.6477 | 0.6031 |
| FLUX.1[dev] | 0.7516 | 0.4887 | 0.6374 | 0.6259 |
| FLUX.1[dev] + `ArtAug` | **0.7541** | **0.5172** | **0.6951** | **0.6555** |

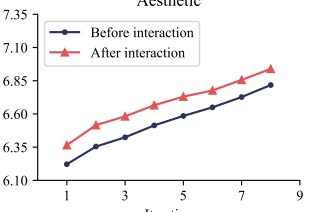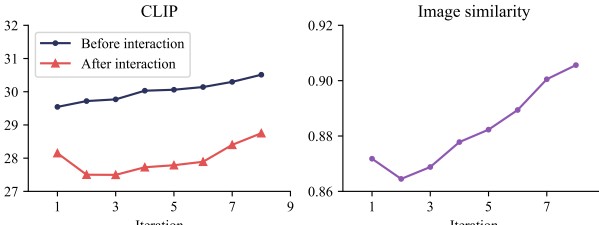

Figure 3: Statistical information of image pairs generated during the iterative training process.

iterations progress, the enhancement effect of the interaction algorithm on image quality diminishes. In the eighth iteration, we are unable to obtain sufficient image pairs for training after filtering, and thus, we stop the training process.

### 4.3 ABLATION STUDIES

We also investigate the effectiveness of the differential LoRA training mentioned in Section 3.3. We compare it with a LoRA model that was trained naively using the enhanced images in the filtered dataset. We evaluate the two training methods in the first iteration of FLUX.1[dev]. By employing the same learning rate and number of training steps, we calculate the basic image quality metrics of the LoRA models. When we naively train the LoRA models, the Aesthetic and CLIP scores are 6.25 and 29.34, respectively; whereas when we employ differential training, the Aesthetic and CLIP scores improve to 6.35 and 29.71, respectively. It is evident that naive LoRA training leads to significant overfitting, resulting in a noticeable decline in text-image alignment, thereby compromising the model's original generative capabilities. Conversely, differential LoRA training better captures the difference in image pairs and avoids overfitting.

The multimodal LLM in the understanding module is crucial in the interaction algorithm. We further compared the performance of Qwen2-VL-72B and other multimodal LLMs. The experimental results are detailed in Appendix B.2. The task of image refinement necessitates not only accurate visual grounding abilities but also a sophisticated understanding of aesthetic principles. We observe that Qwen2-VL-72B can produce more reliable results compared to other models, which is critical for the synthesis of high-quality data. Our approach is agnostic to any specific image understanding model. As more powerful image understanding models emerge, they will provide stronger guidance for image generation, thereby enhancing `ArtAug`'s performance.

## 5 CONCLUSION

In this paper, we explore a method to enhance text-to-image models. To guide these models in generating high-quality images that align with human preferences, we introduce `ArtAug`. `ArtAug` presents a new path for generative model alignment that is not only effective but also highly scalable. By leveraging the synthesis-understanding loop, we transform the problem of aesthetic enhancement from a human-labor-intensive task to a machine-computation-centric one. Based on advanced text-to-image models, we trained `ArtAug` modules in the form of LoRA. Experimental results highlight the substantial improvements achieved through `ArtAug`. Our approach effectively attains alignment training with minimal human resource costs.

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

# A  INTERACTION ALGORITHM

## A.1  PSEUDO CODE

---
**Algorithm 1** Interactive image synthesis algorithm

---
1: **Input:** image prompt $P$, generation module $\hat{\epsilon}_\theta$, understanding module $u$, time steps $T$.
2: Sample $h_0 \sim \mathcal{N}(O, I)$
3: // Generate an image.
4: $h \leftarrow h_0$
5: **for** $t = T$ **to** 1 **do**
6: $\quad \hat{\epsilon} \leftarrow \hat{\epsilon}_\theta(P, t, h)$
7: $\quad h \leftarrow h + (\sigma_{t-1} - \sigma_t)\hat{\epsilon}$
8: **end for**
9: Decode latent representation $h$ to image $X$
10: // Produce modification suggestions.
11: $\{(P_i, M_i)\}_{i=1}^n \leftarrow u(X)$
12: // Regenerate the enhanced image.
13: $h \leftarrow h_0$
14: **for** $t = T$ **to** 1 **do**
15: $\quad \hat{\epsilon} \leftarrow \hat{\epsilon}_\theta(P, t, h)$
16: $\quad \omega = I$
17: $\quad$ **for** $i = 1$ **to** $n$ **do**
18: $\quad\quad \hat{\epsilon} \leftarrow \hat{\epsilon} + \hat{\epsilon}_\theta(P_i, t, h) \cdot M_i$
19: $\quad\quad \omega \leftarrow \omega + M_i$
20: $\quad$ **end for**
21: $\quad h \leftarrow h + (\sigma_{t-1} - \sigma_t)\frac{\hat{\epsilon}}{\omega}$
22: **end for**
23: Decode latent representation $h$ to image $X'$
24: **Return:** image pair $(X, X')$

---

The pseudo code of the interactive image synthesis algorithm is presented in Algorithm 1. For simplicity, this pseudo code uses a flow match-based diffusion model (Liu et al., 2022) as an example, where the parameters $\{\sigma_t\}_{t=1}^T$ is the hyperparameters representing the noise level at each step. This algorithm can be easily extended to other kinds of diffusion models.

## A.2  COMPARED WITH NAIVE PROMPT REFINING

We compared the interaction algorithm outlined in Section 3.1 with the naive prompt refinement approach, as demonstrated in Figure 4. In the naive prompt refinement approach, we leverage the multimodal LLM to directly generate a detailed prompt for the image generation model. This example reveals that regenerating images using only refined prompts typically results in a complete alteration of the scene's overall composition. Conversely, our interaction algorithm is capable of enhancing the details while preserving the fundamental composition, exemplified by the flowers and light. This suggests that our interactive algorithm can ensure the consistency in image content. Therefore, the image pairs generated using the interaction algorithm are better suited for the subsequent differential training process, which is aimed at learning the differences between two images.

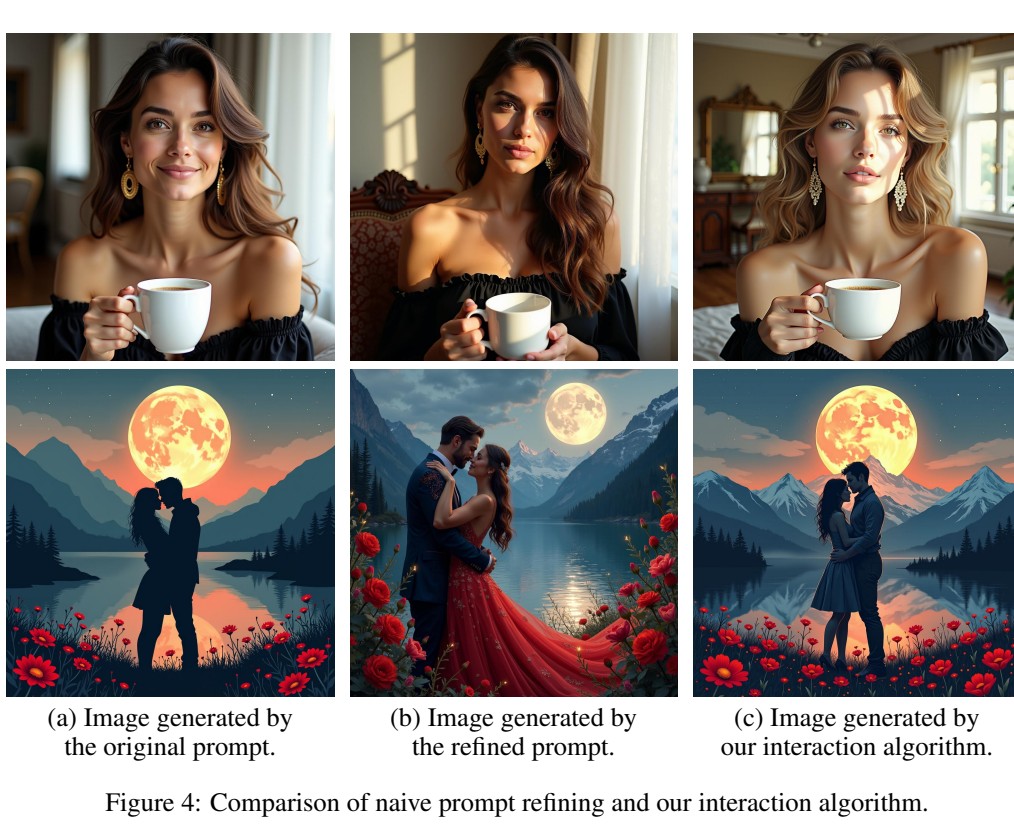

|  |  |  |
|---|---|---|
| (a) Image generated by the original prompt. | (b) Image generated by the refined prompt. | (c) Image generated by our interaction algorithm. |

Figure 4: Comparison of naive prompt refining and our interaction algorithm.

### A.3 EXAMPLES OF INTERACTIONS

Some image pairs generated by our proposed interaction algorithm are displayed in Figures 5 and 6. These images clearly demonstrate that the multimodal LLM can enhance image quality across various fundamental aesthetic aspects. The improvements in the basic aesthetic aspects include:

- **Lighting**: Optimizes the effects of natural and artificial light, ensuring a balance of highlights and shadows.
- **Detail**: Enhances subtle yet crucial elements of objects in the image, boosting realism and visual appeal.
- **Composition**: Adjusts the relative positions of objects within the image, enhancing compositional effects and achieving balanced spatial aesthetics.
- **Ambiance**: Optimizes the background and atmosphere of the image, creating an environment and mood that matches the theme.
- **Clarity**: Improves overall clarity, reducing noise and blur.
- **Color**: Adjusts temperature, saturation, and more, resulting in vibrant, harmonious colors while retaining the original scene's atmosphere.

Beyond these fundamental aesthetic enhancements, our algorithm achieves more advanced effects, including but not limited to:

- **Particle Effects**: Introduces dynamic or special effects, such as particle effects, to images.
- **Shooting Angle**: Alters camera angles for a richer visual experience.
- **Exposure Compensation**: Simulates realistic scenarios like a galaxy appearing with increased exposure.
- **Style Adjustment**: Converts images to specific artistic styles to make them aesthetically pleasing.

- **Background Blur**: Highlights main subjects while ensuring natural transitions in the background.
- **Color Gradient**: Employs color gradients to smoothly transition between colors, resulting in a softer and more harmonious image.

These improvements highlight how well multimodal LLMs can enhance image aesthetics and adapt content and style to suit human preferences. The interactive algorithm effectively transfers the multimodal LLMs' understanding of aesthetics to the text-to-image model, thereby guiding the image generation process.

## B  MULTIMODAL LLMS

### B.1  PROMPT OF MULTIMODAL LLMS

The prompt used in Qwen2-VL-72B has undergone several iterations and extensive testing to ensure its effectiveness in guiding the model to generate enriched and aesthetically pleasing details in the image. This prompt is detailed as follows, where "__prompt__" denotes the original prompt of the text-to-image model.

```
You are a helpful assistant. Given the image please analyze the
    following image and complete the following tasks:

1. Add more details to this image. For example, beautiful light
    and shadow, exquisite decorations, gorgeous clothing,
    beautiful natural landscapes, etc. Caution:
    The added details should be consistent with the original
        description: __prompt__
2. Mark the locations where these details can be added. Caution:
    Each entity should have only a bounding box in the format [x1,
        y1, x2, y2] represented using absolute pixel coordinates.
3. For each bounding box, imagine that we modify it into something
    extremely aesthetically pleasing. Please describe the image
    content of this part using words. Do not use 'should'. Just
    describe it. The aesthetical description should be long.

Please provide the results in JSON format as follows, which can be
    directly loaded by json.loads() in Python:
[
    {
        "bbox": [x1, y1, x2, y2],
        "aesthetical description": "..."
    },
    {
        "bbox": [x1, y1, x2, y2],
        "aesthetical description": "..."
    },
    ...
]
```

### B.2  COMPARISON OF DIFFERENT MULTIMODAL LLMS

In the interactive algorithms, the understanding model is implemented using a multimodal LLM. For the selection of this model, we conduct a comparative analysis. We test multiple multimodal LLMs using the prompts mentioned in Appendix B.1. The models include Qwen2-VL-7B (Wang et al., 2024a), LLaMa3.2-vision-90b-instruct (Dubey et al., 2024), InternVL2-26B (Chen et al., 2024b), Gemini-exp-1206 (Team et al., 2023), Claude-3.5-sonnet (Anthropic, 2025), and Qwen2-VL-72B (Wang et al., 2024a). The locations that each model indicates needed modification are shown in Figure 7, with the corresponding modification prompts described as follows:

**Qwen2-VL-7B**:

1. The couple is beautifully illuminated by the warm glow of the full moon, casting a soft light on their silhouettes.

2. The mountains in the background are adorned with a gradient of colors, transitioning from deep blues at the base to warm oranges near the peaks, creating a stunning contrast against the night sky.

3. The serene lake reflects the moon's light, creating a mirror-like surface that enhances the tranquility of the scene.

4. The trees along the shore are elegantly lit, their dark outlines highlighted by the moonlight, adding depth and dimension to the landscape.

5. The red flowers surrounding the couple add vibrant pops of color, contrasting beautifully with the cool tones of the night sky and enhancing the overall romantic atmosphere of the scene.

**LLaMa3.2-vision-90b-instruct**:

1. The couple's embrace is surrounded by a halo of soft, golden light that illuminates their faces and casts a warm glow on the surrounding landscape. The light is gentle and romantic, highlighting the love and affection between the two people.

2. The full moon in the background is a deep shade of orange, with a subtle gradient effect that gives it a sense of depth and dimensionality. The moon's surface is textured with craters and mountains, adding to its natural beauty and realism.

3. The mountains in the background are covered in a thick layer of mist, which adds to their mystery and allure. The mist is depicted in shades of blue and purple, with touches of pink and orange where it meets the moonlight. The mountains themselves are rugged and rocky, with steep peaks and valleys that add to their dramatic beauty.

4. The lake in the foreground is calm and peaceful, reflecting the beauty of the surrounding landscape like a mirror. The water is depicted in shades of blue and green, with touches of purple and pink where it meets the moonlight. The lake's surface is dotted with ripples and waves, adding to its sense of movement and life.

5. The red flowers in the foreground are vibrant and colorful, adding a pop of color to the otherwise muted palette of the image. The flowers are depicted in various stages of bloom, with some fully open and others still budding. They are arranged in a random pattern, adding to their natural beauty and spontaneity.

**InternVL2-26B**:

1. A luminous full moon with a detailed cratered surface, radiating a warm glow, casting a golden light across the scene. The stars twinkle softly in the night sky.

2. Mountains with jagged peaks, their dark outlines contrasting against the twilight sky, casting deep blue shadows that blend into the cool night.

3. A serene lake reflecting the full moon, mirroring the silhouettes of the mountains and adding a touch of tranquility with its still waters.

4. A field of vibrant red flowers, their petals rich and lush, with leaves that add green highlights, creating a striking contrast to the dark silhouettes of the couple.

5. A dense cluster of coniferous trees, their dark, spiky branches reaching upwards, standing still in the night breeze.

6. The couple's outlines are refined with fluid lines, their forms bathed in the moonlight, emphasizing the romantic embrace.

**Gemini-exp-1206**:

1. The full moon is a radiant celestial orb, its surface a tapestry of soft, glowing craters and valleys. A gentle, ethereal light emanates from it, casting a silvery sheen over the entire

scene. The moon's glow is subtly diffused by a thin veil of high-altitude clouds, creating a halo effect that adds a mystical quality to the night sky. The light reflects off the clouds, painting them with hues of soft orange and dusky pink, creating a breathtaking backdrop for the romantic scene below.

2. The tranquil lake is a mirror of serenity, its surface as smooth as glass, reflecting the moonlit sky and the surrounding landscape with perfect clarity. The water is a deep, rich blue, tinged with hints of emerald green near the edges. Gentle ripples, caused by a soft breeze, create a mesmerizing dance of light and shadow on the water's surface. The reflection of the full moon on the lake is not just a mere image but a shimmering pathway of light that seems to lead into another world, adding a touch of magic and wonder to the scene.

3. The couple stands in a tender embrace, their silhouettes outlined against the luminous backdrop. The woman's dress is a flowing, elegant garment made of a soft, shimmering fabric that catches the moonlight, making her appear almost ethereal. Delicate lace details are visible on the sleeves and hem, adding a touch of romantic sophistication. The man is dressed in a finely tailored suit, the fabric a deep, rich color that contrasts beautifully with the woman's dress. The suit's subtle sheen hints at luxury and refinement, and the way it drapes over his form suggests both strength and tenderness. Their pose is intimate and loving, a perfect representation of a romantic moment frozen in time.

4. The field of red flowers is a vibrant tapestry of color and life. Each flower is a perfect specimen, with velvety petals in a deep, passionate red. Delicate dewdrops cling to the petals, catching the moonlight and sparkling like tiny diamonds. The flowers sway gently in the breeze, creating a sense of movement and vitality. Among the red flowers, there are interspersed tiny wildflowers in shades of white and yellow, adding a touch of delicate beauty and contrast to the scene. The fragrance of the flowers is subtly hinted at, a sweet, intoxicating scent that adds another layer of sensory richness to the romantic atmosphere.

5. The mountains rise majestically in the background, their peaks shrouded in a soft, ethereal mist that glows faintly in the moonlight. The slopes are covered in a lush forest of pine trees, their dark green needles providing a striking contrast to the lighter hues of the sky and the lake. The trees are rendered in exquisite detail, with each branch and needle clearly defined, creating a sense of depth and texture. The play of light and shadow on the mountains is intricate and captivating, with the moonlight highlighting certain areas while others remain in deep shadow, adding a sense of mystery and grandeur to the landscape.

**Claude-3.5-sonnet**:

1. The night sky is a masterpiece of celestial beauty. Thousands of twinkling stars dot the deep blue-green expanse, creating a mesmerizing tapestry of light. Wispy, ethereal clouds float gracefully across the horizon, their edges tinged with a soft, warm glow from the moon's radiance. The stars seem to dance and shimmer, their light pulsating gently in the vast cosmic canvas, inviting the viewer to lose themselves in the infinite wonders of the universe.

2. The full moon dominates the scene, a breathtaking orb of celestial magnificence. Its surface is a intricate map of light and shadow, with every crater and maria visible in stunning detail. The moon glows with an intense golden-orange hue, casting a warm, romantic light across the landscape. Its radiance is so powerful that it seems to pulse with an otherworldly energy, bathing everything below in its enchanting luminescence. The moon's reflection on the lake below creates a perfect, unbroken circle of light, as if two cosmic entities are reaching out to touch one another across the expanse of sky and water.

3. The mountain range is a majestic spectacle of nature's grandeur. Each peak is intricately detailed, with rugged cliffs, deep crevices, and snow-capped summits that catch the moonlight and glisten like diamonds. The silhouettes of the mountains create a dramatic interplay of light and shadow, with each ridge and valley clearly defined. The gradual transition of colors from the deep blue-green of the closer mountains to the misty purple of the distant peaks creates a sense of depth and mystery, inviting the eye to explore every contour and crag of this awe-inspiring landscape.

4. The serene lake is a mirror of tranquility, its surface a flawless reflection of the sky above. The water is so still that every detail of the surrounding landscape is perfectly replicated,

creating a symmetrical masterpiece. Gentle ripples near the shore catch the moonlight, transforming into liquid gold. The lake's edge is adorned with smooth, dark stones that seem to glow with an inner light, their surfaces slick with the kiss of water. The perfect reflection creates an illusion of an infinite world, where sky and water merge into one continuous realm of beauty.

5. The field of red flowers is a vibrant explosion of color and life. Each blossom is exquisitely detailed, with velvety petals that catch the moonlight and seem to glow from within. The flowers sway gently in a soft breeze, creating a hypnotic dance of shadow and light. Delicate stems and leaves are visible among the blooms, adding depth and texture to the floral carpet. Dew drops cling to the petals, sparkling like tiny diamonds in the moonlight. The rich, deep red of the flowers contrasts beautifully with the cool tones of the surrounding landscape, creating a visual focal point that draws the eye and warms the heart.

6. The silhouette of the embracing couple is a poignant symbol of love and connection. Their forms are perfectly balanced, creating a harmonious shape against the backdrop of the moon. The outline of their bodies is crisp and clear, with every detail of their posture and the gentle tilt of their heads towards each other visible. The way they hold each other speaks volumes about their intimacy and affection. Their silhouette creates a powerful contrast against the bright background, making them appear as if they are the sole inhabitants of this magical world, lost in their own private moment of romance and wonder.

**Qwen2-VL-72B**:

1. The couple is dressed in elegant, flowing garments that shimmer under the moonlight. The man's suit is tailored to perfection, with intricate embroidery on the lapels, while the woman's dress is adorned with delicate lace and sequins that catch the light. Their embrace is tender and intimate, creating a sense of warmth and connection.

2. The mountains in the background are majestic and grand, with snow-capped peaks that glisten in the moonlight. The slopes are covered in lush, green forests that add depth and texture to the scene. The sky above is a canvas of stars, with the full moon casting a soft, ethereal glow over the landscape.

3. The serene lake reflects the full moon and the surrounding mountains, creating a mirror-like surface that adds tranquility to the scene. The water is calm and still, with gentle ripples that dance in the moonlight. The reflections of the moon and the mountains create a mesmerizing effect, enhancing the romantic atmosphere.

4. The red flowers surrounding the couple are vibrant and lush, with petals that seem to glow in the moonlight. They are arranged in a natural, wildflower style, adding a touch of whimsy and romance to the scene. The flowers are interspersed with delicate greenery and small, twinkling lights that mimic the stars in the sky.

This task not only requires the model to understand image content but also demands precise visual grounding abilities. Consequently, we observed that most multimodal LLMs are unable to provide viable modification suggestions. Among these models, only Claude-3.5-sonnet and Qwen2-VL-72B meet our requirements. Given that Qwen2-VL-72B is an open-source model, we ultimately decided to use it to construct our understanding module.

## C   HUMAN EVALUATION DETAILS

During evaluation, we used minimal instructions ("Which image looks better?") to prioritize participants' first impressions. Simplified instructions ensure efficient decision-making that better reflects human aesthetic intuition. Each participant was asked to pick an image with better visual quality in 100 image pairs. The user interface provided to participants is shown in Figure 8, with the text displayed in Chinese due to the participants' language background.

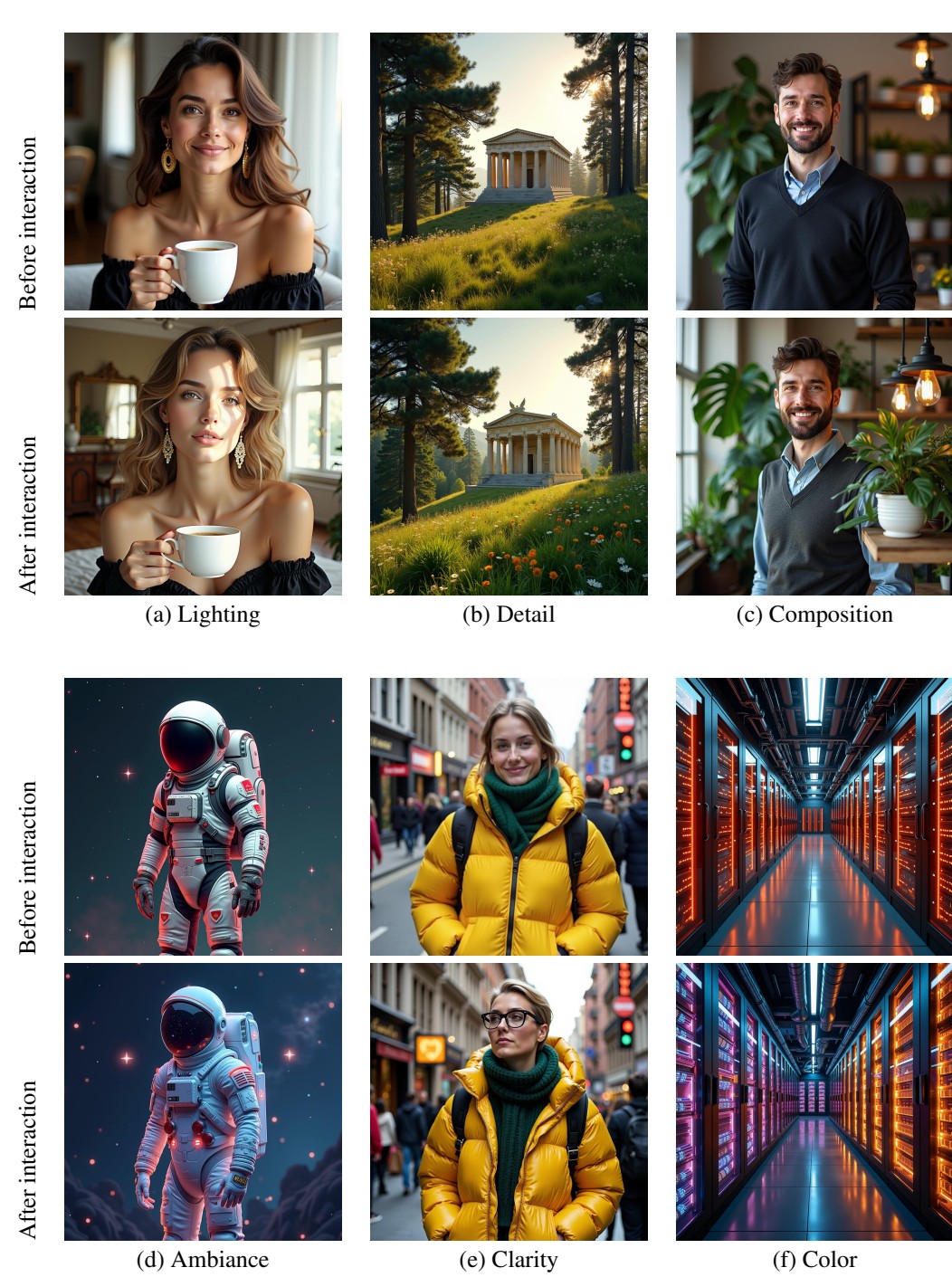

Figure 5: Image examples improved by the interaction algorithm. The enhanced images exhibit aesthetic improvements in various aspects.

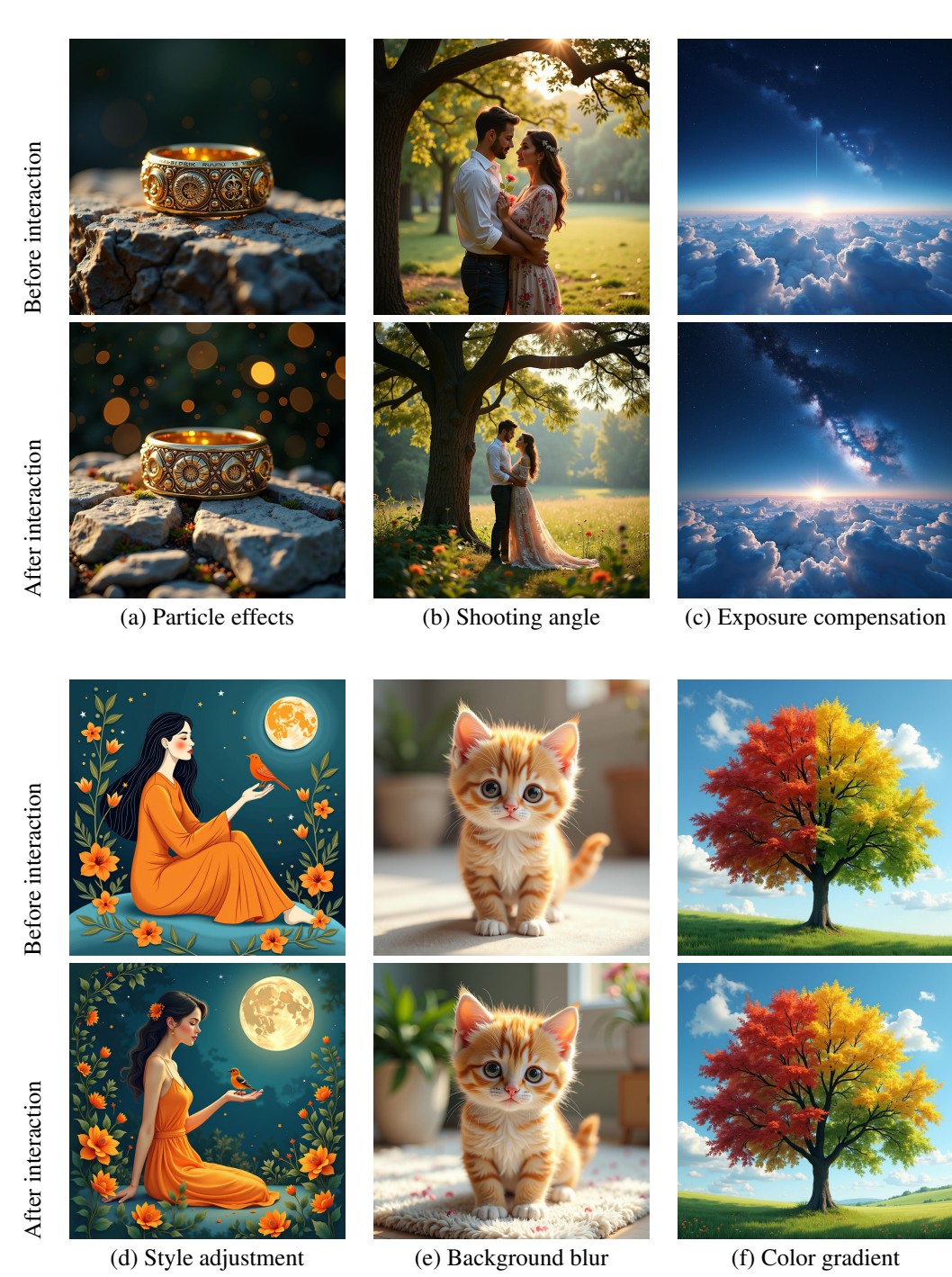

Figure 6: Image examples improved by the interaction algorithm. The multimodal LLM in the interaction algorithm is capable of understanding human preferences and making fine-grained adjustments to the content of images.

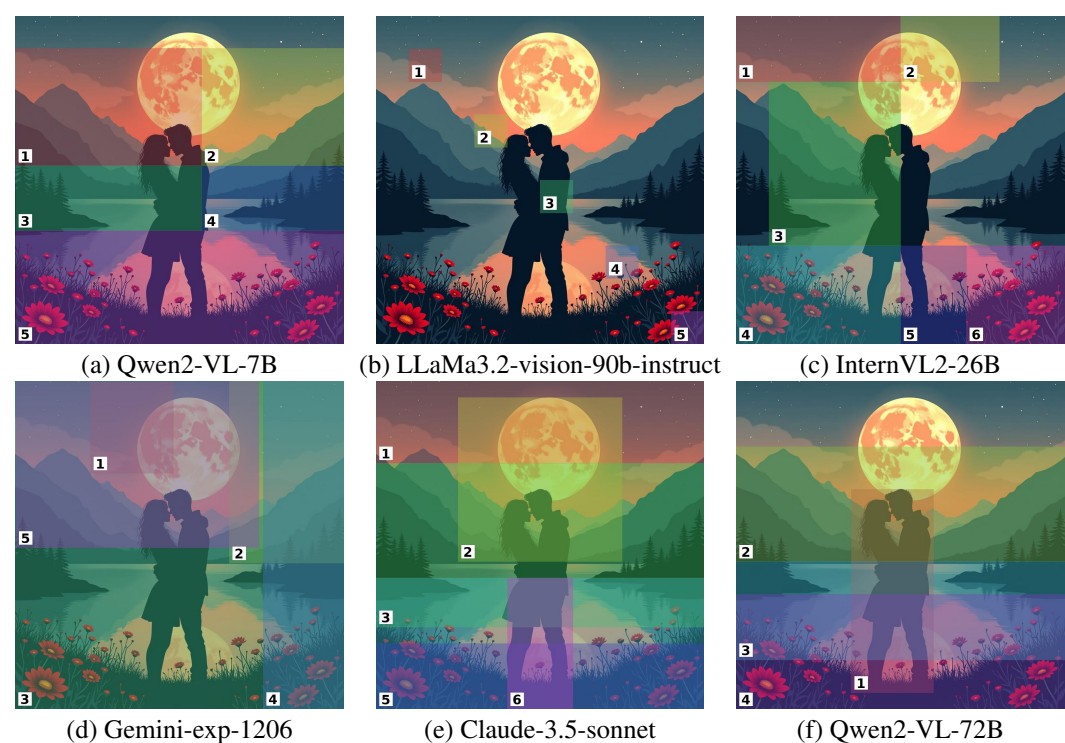

(a) Qwen2-VL-7B  (b) LLaMa3.2-vision-90b-instruct  (c) InternVL2-26B

(d) Gemini-exp-1206  (e) Claude-3.5-sonnet  (f) Qwen2-VL-72B

Figure 7: Comparison of image understanding and refining capabilities of multimodal LLMs. Bounding boxes indicate areas requiring modification, with related prompts provided in the Appendix B.2.

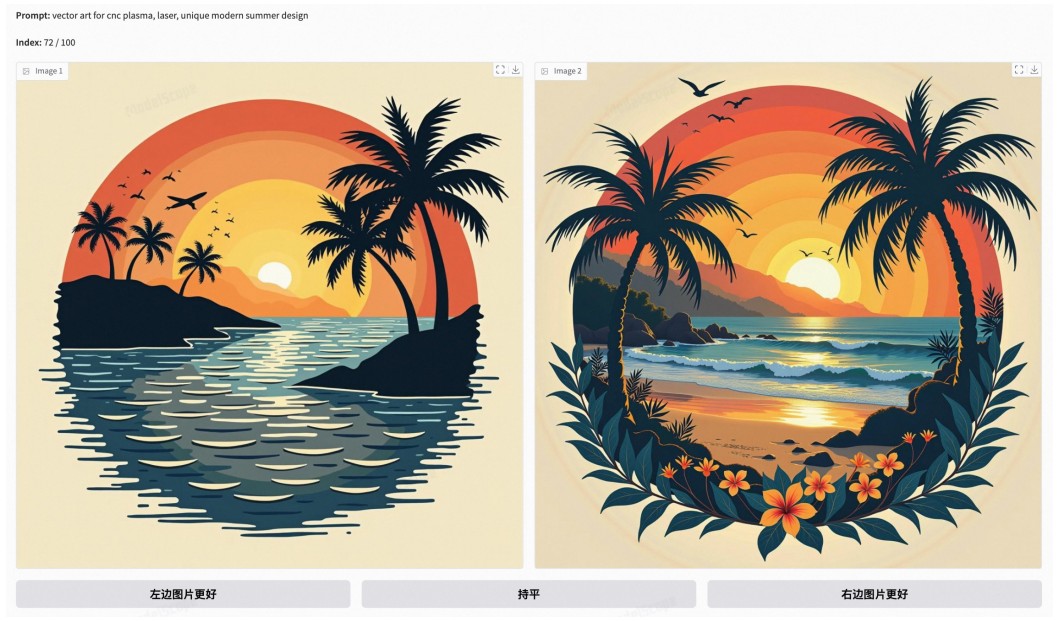

Figure 8: User interface provided to participants in human evaluation.

