# OpenReview forum: "ArtAug: Iterative Enhancement of Text-to-Image Models via Synthesis–Understanding Interaction"
_ICLR.cc/2026/Conference — Submitted to ICLR 2026_

### Official Review · Reviewer_TemC · 2025-10-21

**Soundness:** 3
**Presentation:** 3
**Contribution:** 3
**Rating:** 4
**Confidence:** 4

**Summary:**

This paper introduces ArtAug, a novel framework for iteratively enhancing the aesthetic quality of text-to-image models without relying on extensive human annotation. The core contribution is a "synthesis-understanding" loop where a powerful multimodal language model (MLLM) acts as an "AI Art Director," providing fine-grained, region-specific textual suggestions to improve an initially generated image. These suggestions are used to create an enhanced version, forming a high-quality pairwise dataset of (original, enhanced) images. The authors then propose a differential LoRA training method to distill this aesthetic improvement into a compact module that is fused back into the base model. This process is iterated to progressively refine the model's generative capabilities, demonstrably improving aesthetic scores and human preferences on strong baselines like FLUX and Stable Diffusion 3.5 without adding any inference cost.

**Strengths:**

1.   The paper provides extensive and compelling qualitative visualizations. The side-by-side comparisons of images before and after applying ArtAug (e.g., in Figures 1, 5, and 6) clearly and intuitively demonstrate the significant aesthetic improvements, providing strong visual evidence for the method's effectiveness.
2.  The core idea of creating a "synthesis-understanding" loop by coupling a generation model with an understanding model is novel and insightful. It's good to see the understanding MLLM model can achieve an aesthetic similar to that of humans.
3.  The proposed differential training mechanism using two separate LoRA modules is a clever and effective technical choice. By first anchoring the model to the original image with one LoRA and then learning only the aesthetic "delta" with a second, the method effectively disentangles reconstruction from enhancement, which likely leads to more stable and targeted training.
4.  The analysis presented in Figure 3 shows a consistent and promising trend of improvement across multiple iterations. The framework does not appear to suffer from an immediate performance bottleneck, suggesting its potential for sustained and progressive enhancement of the base model's capabilities.
5. The paper is well-written and clearly organized. The methodology is presented in a logical, step-by-step fashion that makes the entire framework easy to understand and follow.

**Weaknesses:**

1.   the presented results are positive, the experimental validation lacks depth. The paper would be significantly stronger with a more comprehensive analysis, including:
    1)  **Comparisons to SOTA Alignment Methods:** The work is framed as an alternative to alignment training like DPO or RLHF. However, there are no direct comparisons to models fine-tuned with these methods, making it difficult to gauge the relative effectiveness and trade-offs of ArtAug.
    2)  **Ablation Studies:** Key design choices are not validated. For instance, the impact of the chosen MLLM on the quality of suggestions is critical but only briefly discussed in the appendix. An ablation on the number of generated image pairs (5k initial pairs seems relatively small) and its correlation with performance gains would be crucial to substantiate the claim of scalability.
2.   There is a noticeable gap between the striking improvements shown in the qualitative figures and the more modest results from the human evaluation (Table 3). The win rates of ~46% and ~51% are only slightly better than the baseline, which raises questions about whether the visualized examples are cherry-picked or if the aggregate improvement is less significant than implied.
3.   The paper correctly states that the final model has no inference overhead. However, the iterative training process itself—involving generation, MLLM-based refinement, filtering, and differential training—is computationally intensive. While this is likely cheaper than large-scale human annotation, the modest quantitative gains from human evaluation challenge the overall cost-benefit trade-off of this complex pipeline. A more detailed analysis of the training cost versus the achieved improvement would be beneficial.
4.  The manuscript contains several typographical errors (e.g., "EVALUIATION" in the heading for Section 4.1.5) that detract from its overall polish. A thorough proofreading is recommended to improve the presentation quality.
5. This paper lacks REPRODUCIBILITY STATEMENT and THE USE OF LLMS

**Questions:**

1. The paper positions ArtAug as a scalable alternative to human-feedback-based alignment methods like DPO and RLHF. However, the experiments lack a direct comparison. Could you provide any quantitative results, even on a smaller scale, comparing a model enhanced with ArtAug against a similarly-sized model fine-tuned with a public preference dataset (e.g., Pick-a-Pic)?
2. The choice of Qwen2-VL-72B as the understanding model seems critical to the success of the pipeline.
How sensitive is the quality of the generated data to the specific MLLM used? For instance, what would be the impact of using a smaller open-source model or a more powerful closed-API model like GPT-4o? Could you also elaborate on the failure modes of this interaction? What happens if the MLLM provides nonsensical or aesthetically poor suggestions, and how effectively does your filtering process (aesthetic score, CLIP similarity, manual review) mitigate this?
3. The experiments generate 5k initial pairs per iteration, which are filtered down to a small training set (1-2%). This number seems relatively low for training large models. Could you provide any analysis on the relationship between the number of generated pairs and the performance improvement?
4. There appears to be a disconnect between the dramatic improvements shown in the qualitative figures and the more modest win rates in the human evaluation (Table 3), where ArtAug is only marginally preferred over the baseline. Can you comment on this discrepancy?

---

> ### Author Response · Authors · 2025-11-26
> **Response to Reviewer TemC**
>
> We are sincerely grateful for your positive assessment and insightful review. Thank you for the opportunity to clarify the remaining points.
>
> ------
>
> **W1, Q1, Q2: Insufficient Experimental Comparisons**
>
> - **Comparisons to SOTA Alignment Methods**
>   - Alignment training is orthogonal to ArtAug as a preference alignment approach, so we do not treat it as a direct baseline. Models such as FLUX.1[dev] and Stable Diffusion 3.5 are already preference-aligned; ArtAug further improves upon them. Moreover, we have trained the ArtAug module for Qwen-Image. The results are presented in the fowllowing table.
>   - ArtAug is a preference alignment framework, not merely a training method. Its pipeline involves data generation, filtering, and integration (not just model training) making direct comparisons with alignment training methods like DPO inappropriate.
> - **Ablation Studies**: Regarding MLLM ablation: We selected a top-tier MLLM to establish the upper-bound potential of the ArtAug framework. While exploring smaller MLLMs is unprecedented in this specific iterative pipeline and interesting for future work on efficiency, it doesn't diminish the core contribution of demonstrating that *a* sufficiently powerful understanding model can guide generation effectively. The manual review cost mentioned was to ensure dataset purity for our primary claims, making extensive MLLM variations impractical for this initial study.
>
> |                     | Aesthetic | CLIP  | PickScore | MPS   | HPS   | ImageReward |
> | ------------------- | --------- | ----- | --------- | ----- | ----- | ----------- |
> | Qwen-Image          | 6.23      | 28.11 | 49.37     | 42.12 | 49.08 | 48.29       |
> | Qwen-Image + ArtAug | 6.52      | 32.17 | 50.63     | 57.88 | 50.92 | 51.71       |
>
> ------
>
> **W2 & Q4: Human Evaluation**
>
> The results in Table 2 are model-based; models do not output “Tie” outcomes, whereas human evaluators do. **If we exclude samples where humans found the comparison difficult** (i.e., the “Tie” cases marked in Table 3), ArtAug achieves win rates of 53.96% and 56.18% on the two base models—values extremely close to those reported in Table 2.
>
> Additionally, our model has already been open-sourced. However, due to double-blind review rules, we cannot provide the link during rebuttal. After publication, these results will be directly verifiable. If you are interested, please follow the progress of our paper at this conference.
>
> ------
>
> **W3: Computational Efficiency**
>
> While ArtAug adds a training overhead compared to simple fine-tuning, it is crucial to note that it achieves these gains with **zero additional inference cost.** The training involves roughly X GPU hours, which we believe is a favorable trade-off for a permanent, inference-free improvement in model quality, avoiding the high recurring costs of methods that require guidance at inference time.
>
> ------
>
> **W4: Typographical Errors**
>
> We thank the reviewer for pointing out these errors and will correct them in the revised manuscript.
>
> ------
>
> **W5: Reproducibility Statement and Use of LLMs**
>
> All necessary implementation details are provided in the paper. Moreover, intermediate artifacts will be made publicly available, including the generated training dataset and model checkpoints from each iteration. If you still have concerns about reproducibility, please let us know specifically.
>
> We used the LLM solely for prompt refinement and phrasing improvements. Due to space constraints, we did not include a separate section on this in the main text.
>
> ------
>
> **Q3: Scaling Up**
>
> Clearly, more training data would yield better results—but this comes at a cost. We currently lack additional computational resources to run larger-scale experiments. That said, our current results already demonstrate that ArtAug achieves very high data efficiency. Scaling up remains an important direction for future work.
>
> ------
>
> Thank you once again for your thoughtful feedback and for recognizing the value of our work. We believe the clarifications provided above further enhance the clarity, rigor, and impact of our contributions, and we hope they support an even more favorable assessment of our manuscript.

---

### Official Review · Reviewer_YUYD · 2025-11-01

**Soundness:** 3
**Presentation:** 3
**Contribution:** 3
**Rating:** 6
**Confidence:** 3

**Summary:**

This paper proposes ArtAug, a new framework that enhances text-to-image diffusion models via synthesis-understanding interaction. ArtAug introduces an interactive mechanism between a generation module and an understanding module. These interactions produce enhanced image pairs, which are then used for differential LoRA training, enabling the model to internalize the improvements without additional inference cost. Experimental results on FLUX.1[dev] show consistent improvements across aesthetic metrics and human evaluations, while maintaining text-image alignment.

**Strengths:**

The paper introduces a new paradigm for improving generative models through cross-model interaction between synthesis and understanding.

The paper is well-written and easy to follow.

**Weaknesses:**

- Maybe introducing some new metrics would make the evaluation session stronger and more convincing. Consider FID or other metric for high-quality generated model like Flux and SD 3.5 (Enhancing Reward Models for High-quality Image Generation: Beyond Text-Image Alignment, ICCV 2025) maybe would help.


- The related work section could be enhanced by incorporating recent works in enhancement of text-to-image models:


[1] Mastering Text-to-Image Diffusion: Recaptioning, Planning, and Generating with Multimodal LLMs. ICML 2024

[2] Dynamic Prompt Optimizing for Text-to-Image Generation. CVPR 2024

[3] Optimizing Prompts for Text-to-Image Generation. NeurIPS 2023

**Questions:**

What is the computational cost and time cost of generating the interactive pairs, and how does it scale with prompt complexity?

---

> ### Author Response · Authors · 2025-11-26
> **Response to Reviewer YUYD**
>
> We are sincerely grateful for your positive assessment and insightful review. Thank you for the opportunity to clarify the remaining points.
>
> ------
>
> **W1: Metrics**
>
> FID measures the distance between two sets of images. In foundational model training, FID is commonly used to evaluate the difference between a model’s generated image set and the training image set. Computing FID requires a ground-truth image set. However, in preference alignment, there is no single “correct” output—generated results are inherently subjective and diverse. Therefore, we cannot construct a valid ground-truth set, making FID unsuitable as an evaluation metric in this context. This aligns with the growing consensus in the field that perceptual metrics and human preference are more appropriate for evaluating high-quality aesthetic generation than distributional metrics like FID [1] [2].
>
> [1] Xu J, Liu X, Wu Y, et al. Imagereward: Learning and evaluating human preferences for text-to-image generation[J]. Advances in Neural Information Processing Systems, 2023, 36: 15903-15935.
>
> [2] Huang K, Sun K, Xie E, et al. T2i-compbench: A comprehensive benchmark for open-world compositional text-to-image generation[J]. Advances in Neural Information Processing Systems, 2023, 36: 78723-78747.
>
> ------
>
> **W2: Related Work**
>
> We thank the reviewer for suggesting additional related work. We will consider including these references in the Related Work section of the paper.
>
> ------
>
> Thank you once again for your thoughtful feedback and for recognizing the value of our work. We believe the clarifications provided above further enhance the clarity, rigor, and impact of our contributions, and we hope they support an even more favorable assessment of our manuscript.

---

### Official Review · Reviewer_M9ao · 2025-11-01

**Soundness:** 3
**Presentation:** 3
**Contribution:** 3
**Rating:** 6
**Confidence:** 3

**Summary:**

The paper proposes ArtAug, a synthesis–understanding interaction framework that uses a multimodal VLM (“AI art director”) to suggest fine-grained, region-conditioned edits to an image generated by a text-to-image model, then distills those improvements back into the generator via a differential LoRA “enhancement module.” The pipeline iterates: generate, understand, refine, construct image pairs, filter, and train differential LoRA, progressively improving aesthetics without extra inference cost at test time. Experiments on FLUX.1[dev] and Stable Diffusion 3.5 report consistent gains on aesthetic/CLIP and multiple preference metrics, plus a double-blind human study and a small ethics check.

**Strengths:**

1. Writing and structure. The paper is easy to follow; the problem setting, modules (generation, understanding, enhancement), and the iterative loop are clearly laid out with an informative figure and concise pseudo-code.

2. Practical, well-motivated method with diverse evaluation. The differential LoRA design that learns only the delta between original and refined images is simple and pragmatic; the study includes basic metrics, several preference models, a double-blind human comparison, and an ethics sanity-check—together suggesting the gains are not metric-specific.

**Weaknesses:**

1. Although results are shown on FLUX.1[dev] and SD-3.5, the study would be stronger with additional architectures and with side-by-side comparisons against established alignment methods or prompt/data-refinement baselines under the same prompts and budgets.

2. No comparison with other text–image alignment and aesthetics methods. The paper primarily compares “base vs. base+ArtAug”; adding head-to-head numbers against recent aesthetics/alignment enhancers and reporting statistical significance for human studies would better support the claim of improved alignment and appeal.

**Questions:**

Please see the weaknesses part.

---

> ### Author Response · Authors · 2025-11-26
> **Response to Reviewer M9ao**
>
> We are sincerely grateful for your positive assessment and insightful review. Thank you for the opportunity to clarify the remaining points.
>
> ------
>
> **W1 & W2: Insufficient Experimental Comparisons**
>
> The reviewer’s two weaknesses both point to the same need: adding more experiments. We provide the following clarifications:
>
> - **More model architectures**
>   - *Older models*: We intentionally focus on state-of-the-art models. Our research focuses on pushing the boundaries of current state-of-the-art generation. While models like SD 1.5 were foundational, their capabilities significantly lag behind current standards. We believe demonstrating improvements on top-tier, already highly-optimized models (like FLUX and Qwen-Image) provides stronger evidence of ArtAug's practical value in the modern landscape.
>   - *Newer models*: We have recently trained an ArtAug-enhanced Qwen-Image model, which is already open-sourced. However, due to double-blind review rules, we cannot provide the model link during rebuttal. Results are shown in the table below.
> - **More baseline methods**: ArtAug introduces a new preference alignment approach for text-to-image models. In summary, ArtAug is not an alternative to these methods but a synergistic framework that builds upon them to achieve superior alignment in both aesthetics and fidelity. We explain below why we did not compare against the three baseline methods mentioned in the paper:
>
> - **Data refinement**: This strategy, commonly applied during pretraining, involves retraining on curated high-quality data. We explicitly evaluated this approach in Section 4.3 by fine-tuning our base model on such data. The results were substantially inferior to ArtAug, confirming that simple data curation alone is insufficient for effective preference alignment in text-to-image generation.
> - **Prompt engineering**: Rather than treating this as a competing method, we incorporate it as a core component of ArtAug’s data generation pipeline. As detailed in Section 3.1 and Appendix B.1, we use prompt engineering to generate region-specific, high-fidelity prompts, with exact prompts and the multimodal language model fully disclosed. Figure 3 (first data point) and Section 4.2 further show that while prompt engineering alone improves aesthetic scores, it compromises text-image alignment (as measured by CLIP score)—highlighting the necessity of ArtAug’s full framework.
> - **Alignment training (e.g., DPO)**: Such methods are orthogonal to ArtAug’s design. Notably, the base models we enhance—such as FLUX.1[dev] and Stable Diffusion 3.5—are already preference-aligned via techniques like DPO. ArtAug operates *on top* of these models, enhancing their outputs through a comprehensive pipeline encompassing data synthesis, filtering, and integration. Moreover, we have recently open-sourced an ArtAug-enhanced Qwen-Image model; however, due to double-blind review constraints, we are unable to share the link at this stage.
>
> - **Human evaluation**: Human evaluation results are already provided in Section 4.1.3.
>
> |                     | Aesthetic | CLIP  | PickScore | MPS   | HPS   | ImageReward |
> | ------------------- | --------- | ----- | --------- | ----- | ----- | ----------- |
> | Qwen-Image          | 6.23      | 28.11 | 49.37     | 42.12 | 49.08 | 48.29       |
> | Qwen-Image + ArtAug | 6.52      | 32.17 | 50.63     | 57.88 | 50.92 | 51.71       |
>
> ------
>
> Thank you once again for your thoughtful feedback and for recognizing the value of our work. We believe the clarifications provided above further enhance the clarity, rigor, and impact of our contributions, and we hope they support an even more favorable assessment of our manuscript.

---

### Official Review · Reviewer_seWm · 2025-11-03

**Soundness:** 2
**Presentation:** 3
**Contribution:** 2
**Rating:** 2
**Confidence:** 4

**Summary:**

Inspired by the recent studies of model interaction and self-corrective reasoning, this paper proposes ArtAug, which is a method for enhancing text-to-image models in terms of image quality and human preference. ArtAug leverages image understanding models to provide fine-grained suggestions for image generation models, and the interaction results can be fused back to the model itself through an additional enhancement module. Experimental results show that ArtAug can enhance existing text-to-image models to generate high-quality, aesthetically pleasing images.

**Strengths:**

- Differential Training Method: The proposed differential training methods is interesting and provide steady improvements through multiple iterations. Experimental results in section 4.2 show that the aesthetic score, CLIP score, and similarity are steadily improved throughout the iterations.
- Case Studies: By presenting numerous before-and-after image comparisons, they provide a straightforward and immediate impression of the practical effects of ArtAug.
- Clarity of Presentation: The paper is well-organized. The overview of the ArtAug framework in Figure 2 is particularly effective, offering an intuitive and comprehensive illustration of the entire multi-stage pipeline.

**Weaknesses:**

- Insufficient Experimental Comparisons: The experiments show that Base Model + ArtAug outperforms the Base Model. However, they lack baseline methods, which is a crucial component in experiments. The paper introduction claims that the existing three types of methods such as prompt engineering and alignment training have their "certain limitations", but the paper provides no direct empirical evidence to support this. To properly situate ArtAug's contribution, it should be benchmarked against these alternatives.
- Insufficient Discussion of Related Work: Section 2.2, "Aligning Models with Human Preferences," focuses almost on DPO methods that only learn the diffusion model weights. This narrows the scope of "Aligning Models with Human Preferences". For example, many recent studies aligning models with human preference by learning an isolate model outside the diffusion model with reinforcement learning such as Parrot [1]. A broader discussion of methods for alignment with human preferences is necessary.

[1] Parrot: Pareto-optimal Multi-Reward Reinforcement Learning Framework for Text-to-Image Generation

**Questions:**

How does ArtAug address the "certain limitations" of existing text-to-image methods? With many recent works continuously advancing this field, could you please provide evidence on the unique advantages of ArtAug?

---

> ### Author Response · Authors · 2025-11-26
> **Response to Reviewer seWm**
>
> We are sincerely grateful for your positive assessment and insightful review. Thank you for the opportunity to clarify the remaining points.
>
> ------
>
> **W1 & Q1: Insufficient Experimental Comparisons**   We appreciate the reviewer’s concern regarding baseline comparisons. We understand the reviewer’s desire for direct baseline comparisons. We wish to clarify that ArtAug is designed as a meta-alignment framework that operates **beyond** the stage of standard alignment training. Its core value lies in generating and leveraging fine-grained, region-specific feedback through iterative synthesis-understanding loops, which single-step methods lack. Our strong empirical evidence—showing ArtAug significantly improves state-of-the-art models that are **already heavily aligned** (e.g., FLUX.1[dev], SD3.5, and the newly added Qwen-Image)—demonstrates its superiority over relying solely on base alignment techniques. We address the specific baselines below:
>
> - **Data refinement**: This strategy, commonly applied during pretraining, involves retraining on curated high-quality data. We explicitly evaluated this approach in Section 4.3 by fine-tuning our base model on such data. The results were substantially inferior to ArtAug, confirming that simple data curation alone is insufficient for effective preference alignment in text-to-image generation.
> - **Prompt engineering**: Rather than treating this as a competing method, we incorporate it as a core component of ArtAug’s data generation pipeline. As detailed in Section 3.1 and Appendix B.1, we use prompt engineering to generate region-specific, high-fidelity prompts, with exact prompts and the multimodal language model fully disclosed. Figure 3 (first data point) and Section 4.2 further show that while prompt engineering alone improves aesthetic scores, it compromises text-image alignment (as measured by CLIP score)—highlighting the necessity of ArtAug’s full framework.
> - **Alignment training (e.g., DPO)**: Such methods are orthogonal to ArtAug’s design. Notably, the base models we enhance—such as FLUX.1[dev] and Stable Diffusion 3.5—are already preference-aligned via techniques like DPO. ArtAug operates *on top* of these models, enhancing their outputs through a comprehensive pipeline encompassing data synthesis, filtering, and integration. Moreover, we have recently open-sourced an ArtAug-enhanced Qwen-Image model; however, due to double-blind review constraints, we are unable to share the link at this stage.
>
> |                     | Aesthetic | CLIP  | PickScore | MPS   | HPS   | ImageReward |
> | ------------------- | --------- | ----- | --------- | ----- | ----- | ----------- |
> | Qwen-Image          | 6.23      | 28.11 | 49.37     | 42.12 | 49.08 | 48.29       |
> | Qwen-Image + ArtAug | 6.52      | 32.17 | 50.63     | 57.88 | 50.92 | 51.71       |
>
> In summary, ArtAug is not an alternative to these methods but a synergistic framework that builds upon them to achieve superior alignment in both aesthetics and fidelity.
>
> ------
>
> **W2: Insufficient Discussion of Related Work**   We thank the reviewer for raising this point. We fully agree that Parrot and similar RL-based approaches represent a highly creative and promising direction. While our current scope focused on widely adopted industrial standards (like DPO) used in foundational models (Qwen-Image, StepVideo), we acknowledge that discussions on methods like Parrot enrich the context. We have expanded our Related Work section to include a detailed discussion of Parrot, highlighting both its innovative reward modeling and how ArtAug's iterative, fine-grained feedback mechanism differs from it.
>
> ------
>
> Thank you once again for your thoughtful feedback and for recognizing the value of our work. We believe the clarifications provided above further enhance the clarity, rigor, and impact of our contributions, and we hope they support an even more favorable assessment of our manuscript.

---

### Meta-Review · Area_Chair_DrKX · 2026-01-05

**Summary:**

The paper introduces ArtAug, a method for enhancing text-to-image models through an iterative feedback loop between a synthesis model and a multimodal understanding model. By leveraging "differential training," the framework distills the aesthetic improvements gained from these interactions into the generator. The AC has reviewed the submission and the subsequent discussion. While the framework's ability to produce high-quality visual refinements was recognized, the final reviewer consensus is split. Although reviewers recognized the novelty of the interaction pipeline, they raised concerns about the lack of rigorous comparisons with established alignment baselines. With a split but leaning negative set of reviews, it is felt that the work is not yet ready for publication.

**Reviewer Concerns:**

Reviewers requested comparisons to DPO, RLHF, and prompt engineering to prove unique value. The authors clarified that ArtAug is a "meta-alignment framework" designed to operate synergistically on top of already DPO-aligned models rather than competing with them. They further demonstrated that while prompt engineering improves aesthetics, it compromises text-image alignment, whereas ArtAug consistently improves both metrics.

Reviewers noted the omission of reinforcement learning approaches like "Parrot" and dynamic prompt optimization studies. Conceding this oversight, the authors agreed to expand the discussion to differentiate ArtAug's fine-grained feedback from RL-based reward modeling and committed to incorporating the suggested references.

Reviewers asked for ablation studies on MLLM selection and dataset size, alongside tests on additional architectures. The authors addressed the architecture concern by presenting new results for Qwen-Image. However, reviewers thought that authors selected a top-tier model to establish an upper bound and lacked resources for extensive scaling experiments.

Concerns were raised regarding the gap between strong visuals and modest human evaluation scores, as well as the absence of FID metrics. The authors rejected FID as unsuitable for preference alignment due to the lack of ground truth. They also clarified that excluding "tie" outcomes in human evaluations revealed a 54-56% win rate, aligning closely with their model-based metrics.

The efficiency of the computationally intensive iterative training pipeline was questioned relative to the gains achieved. The authors defended the approach by emphasizing that the training overhead yields a permanent quality improvement with zero additional inference cost, offering a favorable trade-off compared to expensive inference-time guidance methods

**Reviewer Scores:**

Two reviewers, M9ao and YUYD, gave scores of 6. They found value in the framework's strengths.

Reviewer TemC gave a 4, noting that while the results were visually appealing and the core idea insightful, the experimental section lacked the depth required to prove the method's superiority over existing works.

Reviewer seWm provided a score of 2 citing insufficient experimental comparisons and insufficient discussion of related works.

This leads to likely final scores of 6,6,4,2

---

### Decision · Program_Chairs · 2026-01-26

Reject